# POLE-Mutant Colon Cancer Treated with PD-1 Blockade Showing Clearance of Circulating Tumor DNA and Prolonged Disease-Free Interval

**DOI:** 10.3390/genes14051054

**Published:** 2023-05-08

**Authors:** Mihir Bikhchandani, Farin Amersi, Andrew Hendifar, Alexandra Gangi, Arsen Osipov, Karen Zaghiyan, Katelyn Atkins, May Cho, Francesca Aguirre, Dennis Hazelett, Rocio Alvarez, Lisa Zhou, Megan Hitchins, Jun Gong

**Affiliations:** 1Department of Hematology and Oncology, Kaiser Permanente Los Angeles Medical Center, Los Angeles, CA 90027, USA; 2Department of Surgery, Division of Surgical Oncology, Samuel Oschin Comprehensive Cancer Institute, Cedars Sinai Medical Center, Los Angeles, CA 90048, USA; 3Department of Medicine, Division of Hematology and Oncology, Samuel Oschin Comprehensive Cancer Institute, Cedars Sinai Medical Center, 8700 Beverly Blvd, AC 1042B, Los Angeles, CA 90048, USA; 4Department of Radiation Oncology, Samuel Oschin Comprehensive Cancer Institute, Cedars Sinai Medical Center, Los Angeles, CA 90048, USA; 5Department of Medicine, Division of Hematology and Oncology, University of California Irvine, Irvine, CA 92868, USA; 6Department of Biomedical Sciences, Cedars-Sinai, Los Angeles, CA 90048, USA

**Keywords:** colon cancer, high tumor mutation burden, immunotherapy, checkpoint inhibitor, POLE, MSS

## Abstract

Colon cancer with high microsatellite instability is characterized by a high tumor mutational burden and responds well to immunotherapy. Mutations in polymerase ɛ, a DNA polymerase involved in DNA replication and repair, are also associated with an ultra-mutated phenotype. We describe a case where a patient with POLE-mutated and hypermutated recurrent colon cancer was treated with pembrolizumab. Treatment with immunotherapy in this patient also led to the clearance of circulating tumor DNA (ctDNA). ctDNA is beginning to emerge as a marker for minimal residual disease in many solid malignancies, including colon cancer. Its clearance with treatment suggests that the selection of pembrolizumab on the basis of identifying a POLE mutation on next-generation sequencing may increase disease-free survival in this patient.

## 1. Introduction

Colon cancer is the third-leading cause of cancer-related death in men and women in the United States, with an estimated 153,020 new cases and 52,550 estimated deaths from colorectal cancer in 2023 [1]. Although the majority of colorectal cancer diagnoses occur in patients of older age (65 years and older), the incidence of colorectal cancer is shifting to earlier ages of diagnosis, where colorectal cancer represents the leading cause of death in men under the age of 50 years [1]. Although most diagnoses of colorectal cancer are found with localized disease (about 15–30% present with metastases), up to 50% of these patients initially diagnosed with localized disease develop metastasis [2]. Surgery is definitive for stage I colorectal cancer [3,4]. For stage II–III rectal cancer, neoadjuvant therapy followed by surgery with or without adjuvant therapy has been a widely accepted standard treatment paradigm [3]. In stage III colon cancer and select cases of stage II colon cancer, surgery followed by adjuvant therapy is a widely accepted treatment standard [4]. Systemic therapy is foundational to the treatment of metastatic disease, however, and is influenced by the molecular profile of the cancer. Examples include combining epidermal growth factor (EGFR) inhibitors with standard cytotoxic chemotherapy for patients with KRAS and BRAF wild-type disease [5], treating patients with BRAF V600E-mutated cancer with dual BRAF and EGFR inhibition [6], combining KRAS G12C inhibitors with anti-EGFR therapies in KRAS G12C-mutated colorectal cancer [7], using HER2-directed therapies in HER2-amplified colorectal cancer [7], or treating NTRK fusion-positive colorectal cancers with NTRK inhibitors [8].

A unique molecular subset of metastatic colorectal cancer includes those with microsatellite instability with high (MSI) or mismatch repair deficient (dMMR) tumors, which harbor exquisite sensitivity to immunotherapy [9,10]. The ability to achieve dramatic and durable responses to immune checkpoint inhibitors in MSI colorectal tumors represents breakthroughs that have propelled ongoing efforts to expand the potential for these provocative outcomes in all patients with metastatic colorectal cancer. Unfortunately, although the incidence of MSI in colorectal cancer can approximate 20% in localized disease, the incidence of MSI in metastatic disease is lower where the majority (90–95%) of metastatic colorectal cancer cases are microsatellite stable (MSS) wherein immunotherapy has not been established as a standard systemic therapy [11]. 

Recent efforts have focused on identifying subsets of MSS metastatic colorectal cancer that could derive benefits from immunotherapy or therapeutic strategies to turn the immune-“cold” nature of MSS tumors into immune-“hot” or immune-sensitive tumors, similar to MSI metastatic colorectal cancer. For example, the immune checkpoint blockade has been combined with other standard systemic agents to attempt to render patients with MSS metastatic colorectal cancer sensitive to immunotherapy, such as those with MSI disease [12]. Among those with MSS metastatic colorectal cancer, a high tumor mutation burden may confer sensitivity to immunotherapy as well [13]. Another subset of patients with MSS metastatic colorectal cancer that may be responsive to immunotherapy are those with somatic mutations in polymerase ɛ (POLE). Although germline variants in POLE and polymerase δ (POLD) have been associated with germline variants in mismatch repair genes resulting in MSI tumors (i.e., Lynch Syndrome), somatic mutations in POLE have been detected in 1% of all colorectal cancers and occur only in MSS colorectal cancer (i.e., mutually exclusive of MSI status) [14,15,16]. POLE is a DNA polymerase that is involved in DNA repair through its exonuclease proofreading domain [17]. POLE is involved in leading strand replication, while mismatch repair proteins are involved in DNA mismatch repair downstream of the function of POLE [18]. As such, somatic mutations in POLE represent a mechanism that can lead to highly immunogenic colorectal tumors with an ultra-mutated phenotype that is independent of the mismatch repair status [14,17]. 

In this report, we present a rare case where a patient with MSS colon cancer that carried a somatic POLE mutation and hypermutated phenotype was treated with the programmed cell death 1 (PD-1) receptor inhibitor pembrolizumab. Treatment with pembrolizumab was associated with long-term disease-free survival and undetectable circulating tumor DNA (ctDNA).

## 2. Case Presentation

A 55-year-old male was diagnosed with right-sided, stage IIIB colon cancer after presenting with abdominal discomfort and bloating. He underwent a colonoscopy which showed a large, non-obstructing cecal mass with a biopsy confirming invasive adenocarcinoma with mucinous features. After staging work-up did not show metastatic disease, he underwent a right hemicolectomy with pathology showing a 6.5 cm moderate-poorly differentiated colonic adenocarcinoma invading through the muscularis propria into pericolorectal tissue with 6/28 lymph nodes involved and negative margins (pT3N2a). There was an intact expression of all DNA mismatch repair (MMR) proteins. He was subsequently treated with twelve cycles of adjuvant 5-fluorouracil, leucovorin, and oxaliplatin (FOLFOX). A computed tomography (CT) scan and colonoscopy after completion of adjuvant chemotherapy were negative for malignancy. 

However, the disease recurred within six months of the last cycle of adjuvant FOLFOX, as demonstrated on surveillance CT scans which identified an enlarged periaortic lymph node that was hypermetabolic on the PET scan. He underwent exploratory laparoscopy with lymph node dissection. This path confirmed mucinous metastatic adenocarcinoma morphologically to be consistent with metastasis from a colonic origin. At the time of recurrence, he had detectable ctDNA (0.23 mean tumor molecules/milliliter) based on the Signatera assessment. 

Next-generation sequencing (NGS) was performed on the primary specimen through Caris Life Sciences. It detected a tumor mutational burden of >150 mutations per megabase along with a pathogenic mutation in POLE P436R with a variant allele frequency of 33%. A missense mutation in P436R c.1307C>G was detected in the exonuclease domain of POLE. NGS also confirmed an MSS phenotype. Multiple pathogenic mutations and variants of unknown status were identified, which unsurprisingly corroborated the high tumor mutational burden detected on NGS. Of these, it was notable that pathogenic mutations in BRCA2 and KRAS G13D were detected, while BRAF status, which was the wild type with the ERRB2 (HER2) amplification, was not detected. Germline testing (Invitae 91-gene panel) was negative for any pathogenic sequence variants. 

Subsequently, the patient began pembrolizumab for M1 disease on the basis of a hypermutated phenotype and pathogenic POLE mutation. After six weeks of pembrolizumab (standard dosing at 200 mg every 3 weeks), the patient’s plasma ctDNA levels had cleared and reached negative (Figure 1). Of note, the patient was a non-carcinoembryonic antigen (CEA) producer, and these levels remained low throughout the treatment course. He completed one year of pembrolizumab, by which point the patient had developed immune-related arthralgias prohibiting further treatment. At the time of this report, CT scans continue to show no evidence of disease, while serial plasma ctDNA levels remain undetectable nearly two years after the initial recurrence.

## 3. Discussion

POLE is a DNA polymerase involved in DNA-leading strand synthesis and base excision repair [17]. Mutations in the proofreading exonuclease domain in POLE lead to ultra-mutated colon cancer that is microsatellite stable [17]. It is estimated that pathogenic variants in the exonuclease domain of POLE are found in 2–8% of colorectal cancers [19,20]. The Cancer Genome Atlas Network conducted a genome-scale analysis of 276 colon cancer samples and found that 16% of the colorectal cancers analyzed were hypermutated [21]. Seventy-five percent of these tumors were MSI-high, while the other 25% carried mutations in POLE, suggesting that POLE mutations are mutually exclusive from microsatellite instability but similarly resemble a hypermutated phenotype [21].

The relationship between MSI-high colorectal cancer and its response to immunotherapy is well-established. Immunotherapy has become the recommended upfront treatment for MSI-H colon cancer on the basis of KEYNOTE-177, which showed significantly improved progression-free survival with the use of pembrolizumab compared to chemotherapy in the first-line setting [9]. There was a trend towards an improved overall survival that did not meet statistical significance, likely due to the high crossover rate from chemotherapy to PD-1 therapy. The immune checkpoint inhibitor, nivolumab, either alone or in combination with the CTLA-4 inhibitor ipilimumab, has also been previously approved in patients with MSI metastatic colorectal cancer who have been previously treated with systemic therapies [22,23]. The significant benefit seen with immunotherapy in these cancers also led the National Cancer Comprehensive Network (NCCN) to make a universal recommendation that all new colorectal cancers be tested for MMR or MSI status [3,4]. The promising benefits afforded by immunotherapy for MSI tumors have been confirmed across multiple solid tumors as well, resulting in U.S. Food and Drug Administration (FDA) approvals for immune checkpoint inhibitors such as pembrolizumab in treatment-refractory solid tumors that have MSI [24]. The significance of the recognition that MSI represents an immunotherapy-sensitive molecular subset is two-fold (1) it has led to the first approval of a systemic agent in a tumor-agnostic manner, and (2) it has invigorated research into identifying other molecular subsets that could predict benefits to immunotherapy. 

In contrast to immunotherapy, fluoropyrimidine-based adjuvant chemotherapy is less beneficial for MSI-high colon cancer than for MSS colon cancer [25]. Recent findings from a prospective, randomized controlled FOxTROT trial investigating the role of neoadjuvant chemotherapy for radiologically staged T3-4, N0-2, M0 colon cancer has added further evidence that patients with localized, operable colon cancer derive little benefit from neoadjuvant chemotherapy as well [26]. Instead, neoadjuvant immunotherapy has elicited promising clinical and pathologic complete response rates in both localized colon and rectal cancer subjects [27,28]. The constellation of these findings has really raised provocative questions as to the role of immunotherapy in place of surgery and conventional neoadjuvant or adjuvant therapy approaches in localized colorectal cancer management. Notably, our patient had a very short disease-free interval after their completion of adjuvant FOLFOX, with disease progression within 6 months of completing the last cycle of chemotherapy. This raises the question of whether POLE-mutant colorectal cancer also benefits less from adjuvant chemotherapy, similar to MSI-high colorectal cancer.

Immunotherapy is thought to be beneficial in MSI-high disease because the higher mutational burden leads to a large number of neoantigens in these tumors. Tumor cells upregulate PD-L1 and other immune checkpoints to prevent the host immune system from recognizing neoantigens in the tumor. PD-1 inhibitors reverse the adaptive immune resistance of cancer, allowing host T-cells to recognize the highly mutagenic MSI-high cancer. It is known that compared to MSS colon cancer, colon cancer with mutations in the exonuclease domain in POLE has a higher mutational burden, higher expressions of immune checkpoints such as PD-L1, and a higher expression of T-cell markers [14,29]. We and others have shown that compared to pMMR or MSS colorectal tumors, POLE-mutant colorectal tumors exhibit increased CD8+ lymphocytic infiltration, tumor-infiltrating immune cells with a Th1 phenotype, the expression of cytotoxic T-cell markers, and effector cytokines similar to the degree observed with immunogenic MSI-high colorectal cancers [14,30]. Together, this suggests that POLE-mutated colon cancer may also respond well to the PD-1 blockade.

Our case adds to the growing literature showing POLE-mutant cancers responding to checkpoint inhibition [30,31,32,33,34,35]. Our case, however, is notable for being among a handful of cases of POLE-mutated endometrial and colon cancer that responded dramatically to the immune checkpoint blockade in the context of mutations in the exonuclease proofreading domain of POLE leading to a tumor mutational burden greater than 100 mutations per megabase [32,33,34,35]. The POLE missense mutation seen in our patient, P436R c.1307C>G, is a less commonly seen mutation in POLE and has not been previously described as sensitive to immune checkpoint inhibition. Structural studies have implicated that pathogenic POLE mutations cluster around the active DNA-binding sites of the exonuclease proofreading domain with one mutation, P436R (i.e., the mutation in our patient), which resides in a disordered loop that becomes ordered on DNA binding [36]. POLE mutations that lie peripheral to the exonuclease domain must act indirectly if they are pathogenic [36]. The POLE variant seen in our patient, P436R, has been documented in a handful of endometrial and colon cancer cases and has been described to have a high MutationAssessor-predicted functional impact score (MASS PIFS) and a damaging effect on other predicted functional impact scores [21,37,38]. Specifically, the P436R c.1307C>G missense mutation in POLE has been shown to have a Sorting Intolerant From Tolerant (SIFT) score of 0.01 and a Polymorphism Phenotyping (PPH) v2 score of one, both indicating that this variant is considered to be deleterious rather than benign [37]. As such, we are among the first to present this POLE variant to be associated with an ultra-mutated phenotype in an otherwise MSS colon cancer subject receiving an immune checkpoint blockade.

Another interesting aspect of our case was the change in ctDNA over time. ctDNA are fragments of DNA released by the tumor and are detectable in the bloodstream. In the largest observational cohort to date of postsurgical ctDNA surveillance in resected colorectal cancer, the presence of ctDNA has been independently associated with decreased recurrence-free survival [39]. In a cohort study of 96 consecutive patients diagnosed with stage III colon cancer treated with surgery and adjuvant chemotherapy, the 3-year recurrence-free interval was 30% when ctDNA was detectable compared to 77% when ctDNA was not detected [40].

As ctDNA was not checked in the interval after surgery and before treatment with pembrolizumab in our patient, it is unclear whether the observed clearance of ctDNA was driven by surgery or by pembrolizumab. Nonetheless, ctDNA still remains undetectable more than eighteen months after treatment with pembrolizumab, which is notable, particularly in the context of relapse through M1 disease in our patient. Furthermore, it is noteworthy that ctDNA was a more reliable marker for disease activity in this patient than CEA, which remained low throughout the treatment course due to his non-CEA-producing tumor. Early studies have suggested that plasma ctDNA could serve as a dynamic marker of tumor response to immunotherapy in patients with metastatic colorectal cancer that is both MSS and MSI-high [41]. A large, multicenter, prospective, observational study enrolling over 1500 patients with solid tumors is underway to evaluate the clinical utility of serial plasma ctDNA assessments on clinical decision-making inclusive of the continuation/discontinuation or intensification/de-intensification of immunotherapy in these patients [42]. ctDNA status following metastectomy may be relevant, with multiple randomized controlled trials underway to explore whether tailoring systemic treatment following metastectomy based on ctDNA can improve outcomes [43]. In our patient, =continued undetectable ctDNA suggests that the use of pembrolizumab following metastectomy may be lengthening the patient’s disease-free interval. Our report lastly adds to the literature by being the first case to demonstrate the feasibility of using ctDNA to monitor minimal residual disease in POLE-mutated colorectal cancer.

## 4. Conclusions

Our case highlights the significant therapeutic benefits that may arise from the presence of a pathogenic POLE mutation in a recurrent, metastatic colon cancer subject whose tumor was otherwise MSS. Evidence is growing to suggest the excellent benefits of immune checkpoint inhibition across POLE-mutant cancers when compared to POLE wild-type tumor types. Our case distinguishes itself from others in the literature; however, we present a previously undescribed POLE variant mutation in the exonuclease proofreading domain in a subject with an MSS but ultra-mutated colon cancer treated with pembrolizumab. Further, this is the first known case showing an association between the PD-1 blockade and sustained clearance of ctDNA in POLE mutant colon cancer. Further prospective studies are needed to reproduce the effect of checkpoint inhibition on POLE-mutant colorectal cancers in larger clinical settings. This may lead to more standardized treatment approaches for this actionable molecular feature in colorectal cancer and other tumor types similar to MSI-high tumors.

## Figures and Tables

**Figure 1 genes-14-01054-f001:**
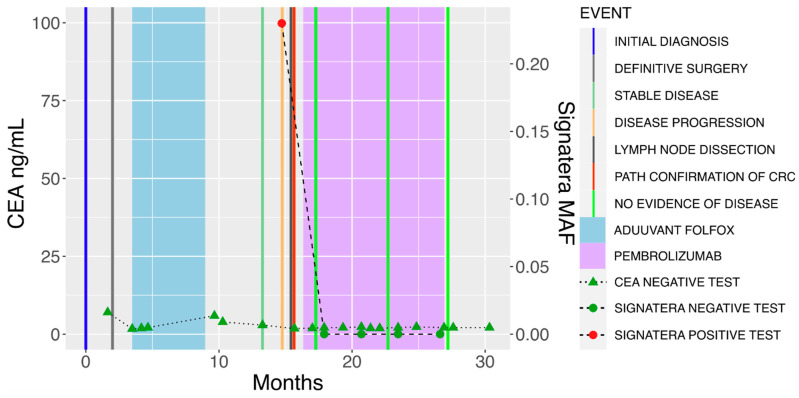
Circulating tumor DNA clearance in POLE-mutant colon cancer treated with pembrolizumab. Radiographic recurrence following treatment with adjuvant FOLFOX for initial stage IIIB colon cancer was corroborated by positivity in plasma-based ctDNA (orange bar). Following resection of a metastatic lymph node confirming colon cancer recurrence (red bar) that was MSS, but POLE mutated on molecular profiling, sustained clearance of ctDNA levels to zero was achieved with pembrolizumab therapy (purple shaded area). Notably, CEA levels were low and uninformative throughout the treatment course for this case.

## Data Availability

Not applicable.

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
