# Peer review of "POLE-Mutant Colon Cancer Treated with PD-1 Blockade Showing Clearance of Circulating Tumor DNA and Prolonged Disease-Free Interval"

_genes, 2023, doi:10.3390/genes14051054_

Round 1

Reviewer 1 Report

The authors reported a case presentation of a colon cancer patient who was effectively treated with pembrolizumab. The colon cancer tissue was analyzed with next-generation sequencing (NGS), detecting high tumor mutational burden (TMB) with POLE P436R. During the treatment in this patient, ctDNA based on Signatera assessment was eliminated.

This manuscript is quite important and requires some modifications.

First, pathogenicity of POLE P436R mutation should be explained. The mutation NM_006231.4(POLE): c.1307C>G (p.Pro436Arg) has not been reported.

Second, are there any histological significance in surgical resected specimen in this high TMB colon cancer. For example, lymphocyte infiltration to the cancer cells may suggest the sensitivity to immune checkpoint inhibitor.

Small points

Detailed results of NGS may be informative to the readers.

Figures can be put together into one figure.

Author Response

We thank the reviewer for their comments and suggestions. We have included a discussion on the pathogenicity of the POLE mutation with references that support that the POLE variant (P436R c.1307C>G) in our case is indeed pathogenic in the Discussion section. We unfortunately do not have histology available from the surgical specimen to evaluate the tumor infiltrating lymphocytes and other immune parameters in our case. We have included in the Discussion, however, evidence from multiple studies that support that POLE mutant colon cancer is characterized by increased CD8+ TILs and other immune parameters that support its responsiveness to immunotherapy, similar to MSI-high colon cancer. We have expanded on the details of the NGS in the Case Presentation. We have combined both figures into one figure, as suggested.

Reviewer 2 Report

This article reports a response to immunotherapy in a CRC patient with Microsatellite stable but POLE mutated tumor.  This is important study that could help other cancer patients with POLE mutated tumors who may get benefited from immunotherapy.

Suggestions:

Shorter time scale on x-axis of Fig 1. would be better to understand the timeline of all events. 

add more details of the immunotherapy protocol, demographic details related to patient if possible.

Author Response

We thank the reviewer for reviewing this manuscript and for the comments and suggestions provided. We have modified the Figure and shortened the X-axis as requested to improve the ease in understanding the timing of events. We have included the dosing regimen (standard dosing) but due to confidentiality reasons are unable to expand more on the demographics of the patient.

Reviewer 3 Report

The authors report on a case of POLE-mutant colon cancer treated with pembrolizumab.

Definitely an interesting topic. However, it appears that all recurrence (in a para-aortic lymph node) was resected, so with an R0 resection the patient was rendered disease free by surgery alone. In this instance the pembrolizumab would better be considered adjuvant therapy and the clearance of ctDNA and prolonged disease free survival could be attributed to surgery alone, with any impact of the IO unknown.

There is a typo in the first author name for ref 4

Author Response

We thank the reviewer for taking the time to review the manuscript and provide feedback. We do agree that it is difficult to ascertain whether it was the surgery or the immunotherapy that rendered the ctDNA levels to zero due to the timepoints of evaluation. We nonetheless do believe that given the rarity of this patient subset who can derive a meaningful benefit to immunotherapy despite being MSS, there is still value to comment on a previously poorly described pathogenic variant in POLE that associates with an ultramutated phenotype conducive to checkpoint blockade. We also believe that ctDNA monitoring may have a role with metastatectomy in mCRC, which our report provides evidence to support this strategy as well. We recognize that whether surgery or immunotherapy resulted in clearing of the ctDNA, the patient met definition for M1 disease and we believe this to be the first case of a POLE mutant colon cancer showing feasibility in serially following ctDNA after receipt of immunotherapy. We have expanded in the manuscript reasons for the novelty and relevance of our findings. We have corrected typos to author references and included additional references to support our claims.

Reviewer 4 Report

This case report is very interesting. It describes the possibility of using immunotherapy (with PD-1 inhibitors) also in cases of colorectal cancer with MSS if mutations in polymerase epsilon (POLE) are present. It is necessary to correct, in the Summary and Introduction, that it is polymerase epsilon whose abbreviation is POLE.

In line 124, "exonucleoside domain" should be corrected to "exonuclease domain".

The weakness of this case report is the absence of data for ctDNA in the period 0 - 250 days.

Author Response

We thank the reviewer for taking the time to review our report and provide suggestions. We have corrected the instances in the manuscript where E should be epsilon for POLE. We have corrected the instance of the misspelling of exonuclease domain. We agree that the earlier ctDNA assessments would have been more supportive to our manuscript, but we do believe that in the setting of M1 disease for our case there is still value to the manuscript as it 1) presents a previously undescribed variant in POLE that has evidence to support its pathogenicity and ultramutated phenotype, 2) highlights the importance of identifying MSS subsets that have immunogenic phenotypes similar to MSI-H, of which POLE represents a rare but clinically relevant subset, 3) supports the role of ctDNA monitoring following metastectomy for M1 colorectal cancer, and 4) illustrates for the first time the feasibility of serially following ctDNA in a POLE mutant colon cancer receiving immunotherapy. We have expanded on the discussion of these points throughout the manuscript to reinforce the novelty and value of our manuscript.